# Revisiting the origins of the *Sobemovirus* genus: A case for ancient origins of plant viruses

**Mahan Ghafari**[1]*, **Merike Sõmera**[2], **Cecilia Sarmiento**[2], **Annette Niehl**[3], **Eugénie Hébrard**[4], **Theocharis Tsoleridis**[5], **Jonathan Ball**[5,6], **Benoît Moury**[7], **Philippe Lemey**[8], **Aris Katzourakis**[1], **Denis Fargette**[4]*

**1** Department of Biology, University of Oxford, Oxford, United Kingdom, **2** Department of Chemistry and Biotechnology, Tallinn University of Technology, Tallinn, Estonia, **3** Julius Kühn Institute (JKI)–Federal Research Centre for Cultivated Plants, Institute for Epidemiology and Pathogen Diagnostics, Braunschweig, Germany, **4** PHIM Plant Health Institute, Univ Montpellier, IRD, CIRAD, INRAE, Institut Agro, Montpellier, France, **5** The Wolfson Centre for Global Virus Research and School of Life Sciences, The University of Nottingham, Queen's Medical Centre, Nottingham, United Kingdom, **6** Department of Tropical Disease Biology, Liverpool School of Tropical Medicine, Pembroke Place, Liverpool, United Kingdom, **7** INRAE, Pathologie Végétale, Montfavet, France, **8** Department of Microbiology, Immunology and Transplantation, Rega Institute, KU Leuven, Leuven, Belgium

* mahan.ghafari@ndm.ox.ac.uk (MG); denis.fargette@ird.fr (DF)

**Data Availability Statement:** All data and software code are available on our GitHub repository (https://github.com/mg878/sobemovirus_genus).

## Abstract

The discrepancy between short- and long-term rate estimates, known as the time-dependent rate phenomenon (TDRP), poses a challenge to extrapolating evolutionary rates over time and reconstructing evolutionary history of viruses. The TDRP reveals a decline in evolutionary rate estimates with the measurement timescale, explained empirically by a power-law rate decay, notably observed in animal and human viruses. A mechanistic evolutionary model, the Prisoner of War (PoW) model, has been proposed to address TDRP in viruses. Although TDRP has been studied in animal viruses, its impact on plant virus evolutionary history remains largely unexplored. Here, we investigated the consequences of TDRP in plant viruses by applying the PoW model to reconstruct the evolutionary history of sobemoviruses, plant pathogens with significant importance due to their impact on agriculture and plant health. Our analysis showed that the *Sobemovirus* genus dates back over four million years, indicating an ancient origin. We found evidence that supports deep host jumps to Poaceae, Fabaceae, and Solanaceae occurring between tens to hundreds of thousand years ago, followed by specialization. Remarkably, the TDRP-corrected evolutionary history of sobemoviruses was extended far beyond previous estimates that had suggested their emergence nearly 9,000 years ago, a time coinciding with the Neolithic period in the Near East. By incorporating sequences collected through metagenomic analyses, the resulting phylogenetic tree showcases increased genetic diversity, reflecting a deep history of sobemovirus species. We identified major radiation events beginning between 4,600 to 2,000 years ago, which aligns with the Neolithic period in various regions, suggesting a period of rapid diversification from then to the present. Our findings make a case for the possibility of deep evolutionary origins of plant viruses.

The code for the PoW model is also available on GitHub (https://github.com/mg878/PoW_model).

**Funding:** Estonian-French science and technology cooperation programme Parrot travel grant 2021-2022 for the project "Emergence and divergence of plant viruses on cereals: Sobemoviruses - a case study" was granted to E.H. and C.S. and contributed to this research. P.L. acknowledges support from the European Union's Horizon 2020 project MOOD (grant agreement no. 874850) and the research and innovation programme (grant agreement no. 725422-ReservoirDOCS), from the Wellcome Trust through project 206298/Z/17/Z, and from the Research Foundation - Flanders ('Fonds voor Wetenschappelijk Onderzoek - Vlaanderen', G0D5117N and G051322N). A.K. acknowledges support from the European Research Council (grant number 101001623-PALVIREVOL). The funders had no role in study design, data collection and analysis, decision to publish, or preparation of the manuscript.

**Competing interests:** The authors have declared that no competing interests exist.

## Author summary

Reconstructing the deep evolutionary history of viruses presents significant challenges, particularly due to a common decline in their evolutionary rate estimates over time. This decline can lead to substantial underestimations of their evolutionary history when divergence time since the most recent common ancestor are estimated based on extrapolations of the short-term evolutionary rates. Our study revisited the evolutionary timeline of the *Sobemovirus* genus, which was previously thought to have originated about 10,000 years ago based on short-term rate extrapolations. By employing the Prisoner of War model, which accounts for changes in virus rate estimates over time, and by analyzing updated datasets of sobemoviruses including metagenomic samples, we found that the *Sobemovirus* genus likely originated over four million years ago. This timeline significantly extends beyond prior estimates, challenging the conventional view of the origins of sobemoviruses and suggesting that they may have ancient beginnings. Furthermore, our study revealed that these viruses have undergone considerable diversification, influenced by agricultural development and environmental changes, particularly over the last 5,000 years. This study paves the way for new insights into the deep evolutionary history of plant viruses and the implications of their long-term interactions with hosts in the context of agriculture.

## Introduction

Reconstructing deep evolutionary history of organisms relies on accurate estimation of their evolutionary rates. However, the discrepancy between short- and long-term evolutionary rate estimates in many organisms, in particular viruses, makes any extrapolation of evolutionary rates based on inference from short timescales challenging [1]. This problem, which later became known as the time-dependent rate phenomenon (TDRP) for viruses, describes a ubiquitous decline in the evolutionary rate estimates with the timescale of measurement. The TDRP is best explained empirically by a power-law rate decay and has been extensively documented in animal and human viruses [2–5]. Several factors can contribute to TDRP such as incomplete purifying selection [6,7], accelerated evolution upon introduction to a new host environment [8,9], and site substitution saturation [2,10]. In addition to these biological processes, several other factors such as misspecification of substitution models can also contribute to time-dependent changes in the evolutionary rate estimates [11].

In a recent work [10], a mechanistic evolutionary model, also known as the Prisoner of War (PoW) model, was proposed that readily explains and corrects for time-dependent rate effects across deep evolutionary timescales and reproduces the empirically observed power-law rate decay found for RNA and DNA viruses. The PoW model operates on the principle that fractions of nucleotide positions evolve at different rates due to factors like epistasis and nucleotide biases. This allows the model to effectively handle the substitution saturation, particularly at those sites that are rapidly evolving, thereby providing a more precise determination of divergence times over extended periods. These timescales are where the impact of site saturation becomes most pronounced, and where standard molecular clock models tend to significantly underestimate the evolutionary timescales [4]. Additionally, the PoW model takes into account the differing saturation points of RNA and DNA viruses, allowing for a flexible reconstruction of the deep evolutionary history of viruses. This model has so far been implemented to reconstruct evolutionary history of many viruses including human and endemic

coronaviruses, foamy viruses, hepaciviruses, human and simian immunodeficiency viruses, monkeypox virus, and variola virus [10,12–17].

TDRP has also been reported in plant viruses [18]. Like many rapidly-evolving animal RNA viruses, a few plant viruses with a robust molecular clock signal based on isolates collected over a few decades have a substitution rate of ca. $10^{-3}$ substitutions per site per year (s/s/y) [19]. By contrast, their substitution rate estimated over a century and longer time periods and based on calibrations through historical events, are typically an order of magnitude smaller, ca. $10^{-4}$ s/s/y [20]. There is additional evidence that points to the overall stability of plant viruses over longer time periods based on reconstruction of archaeological virus genomes [21], integration of endogenous viral elements [22], and virus-host co-divergence [18]. However, these estimates are scarce and possibly obscured by contamination with recent isolates, recombination events, unresolved or poorly dated host phylogenies, and controversial or partial co-evolutionary scenarios. Yet they collectively point to much slower rates of ca. $10^{-4}$–$10^{-8}$ s/s/y, orders of magnitude slower than their short-term rate estimates and closer to that of their plant hosts over longer timescales. Despite these independent lines of evidence in support of time-dependent changes in the evolutionary rate estimates, TDRP has been largely ignored in molecular dating studies of plant viruses. A prominent example includes a study of the reconstruction of the rice yellow mottle virus (RYMV) diversification [23], a virus of the *Sobemovirus* genus (*Solemoviridae* family). This study showed that by extrapolating the short-term substitution rate estimate of RYMV (ca. $10^{-3}$ s/s/y) to much longer timeframes, the diversification of the solemovirids stretches back to the Neolithic's agricultural expansion around 10,000 years ago. Other studies, taking a similar approach (i.e. not accounting for TDRP), have associated plant virus diversification to historical events and periods within the last centuries or few millennia, such as the Neolithic, crop domestication, Post-Columbian exchanges, and trade along the Silk Road in various regions but, crucially, never beyond (see e.g. [20,24]).

Understanding the evolutionary history of sobemoviruses is important as they include plant pathogens with high economic impact, of which RYMV is the most devastating [25]. The *Sobemovirus* genus includes 21 recognized species and 5 candidate species [26], some of which having limited geographical distribution (single continent or country) while others are found worldwide. Many species are transmitted by beetles (order Coleoptera) and a few by aphids (family Aphididae, order Hemiptera) or mirids (family Miridae, order Hemiptera). All sobemoviruses are readily propagated by mechanical wounds. Infections can be asymptomatic or cause severe disease dependent on the virus-host combination and environmental conditions. Symptoms mainly include mosaic and mottle of the infected leaves. While the host range of each virus species in this genus is narrow and confined to a few species of a single plant family, with the exception of sowbane mosaic virus, sobemoviruses collectively infect a wide range of monocot and dicot hosts mainly including plant host species in the Poaceae, Fabaceae, and Solanaceae families [27].

Icosahedral virions of sobemoviruses are 26–32 nm in diameter and composed of 180 monomers of viral capsid protein (CP) on a T = 3 lattice symmetry. Particles contain a single molecule of positive sense single-stranded RNA, about 4.0–4.5 kb in size. A subgenomic RNA (sgRNA) molecule, co-terminal with the 3′ end of the genomic RNA is synthesized in the virus-infected cells. Both genomic and subgenomic RNAs have a viral genome-linked protein (VPg) covalently bound to their 5′ end. The 3′ terminus is non-polyadenylated and does not contain a tRNA-like structure. Several sobemoviruses encapsidate a circular viroid-like satellite RNA (220–390 nt). The genome comprises a polycistronic, positive-sense RNA molecule without a 3′-poly(A) tail. Additionally, it contains five open reading frames (ORFs), and the 5′-end has a covalently attached VPg. The 5′-proximal ORF1 encodes a non-conserved RNA silencing suppressor protein needed for systemic spread, followed by ORFx. Next ORF (ORF2a) encodes

a polyprotein that is expressed by ribosomal leaky scanning and cleaved autocatalytically to different functional subunits (membrane anchor, serine protease, VPg and C-terminal domains). Expression of the viral RNA-dependent RNA polymerase (RdRP) as an alternative C-terminal domain of the polyprotein (from ORF2b) is regulated by a −1 programmed ribosomal frameshift. The 3′-proximal ORF3 of sobemoviruses, expressed from sgRNA, encodes CP [26].

Here, we revisited the evolutionary history of sobemoviruses, taking into account the impact of TDRP on estimating diversification events of sobemovirus ancestry using the PoW model. By providing the first short-term substitution rate estimate of cocksfoot mottle sobemovirus (CfMV), we showed that the rate estimates are largely similar between CfMV and RYMV. We then reconstructed the sobemovirus history using the PoW model and showed that they have a deep evolutionary history that stretches back to millions of years. In particular, we found that the CfMV/RYMV split occurred more than five hundred thousand years ago which falls within the Pleistocene epoch, much earlier than the start of the Neolithic–a period of technical development associated with agriculture that occurred at different periods among regions over the past ten thousand years [28,29]–initially predicted based on extrapolation of short-term rate estimates. We then investigated the sobemovirus radiation, the links between virus lineages and plant host families, and estimated speciation dates using sequences of the polymerase gene of 26 sobemovirus species and 42 additional sequences of the phytobiome defined as 'plants, their environment, and their associated communities of organisms' [30]. Finally, we discussed the implications of our results for the evolutionary history of plant viruses.

## Results

### Short-term substitution rates of CfMV and RYMV

We first assessed and confirmed the strength of temporal signal present in the CfMV dataset through the tip cluster randomization test in root-to-tip regression (see **Methods**). We estimated the short-term substitution rate of the coat protein gene (ORF3) at $7.8 \times 10^{-4}$ s/s/y (95% highest posterior density (HPD): $6.4–9.5 \times 10^{-4}$ s/s/y) for RYMV and at $5.0 \times 10^{-4}$ s/s/y (95% HPD: $2.3–8.2 \times 10^{-4}$ s/s/y) for CfMV using the same substitution model in our Bayesian inference (**Fig 1**). The wider variation in the CfMV posterior rate distribution likely reflects the lower number of sequences and weaker clock signal compared to the RYMV dataset. The inferred substitution rate of RYMV for the polymerase gene was $6.5 (5.2–7.9) \times 10^{-4}$ s/s/y, similar to and slightly lower than that of the coat protein gene. The similarity between the short-term substitution rates of two sobemovirus species supports the choice of the RYMV substitution rate as the representative short-term rate to reconstruct the evolutionary history of the sobemovirus genus using the PoW model.

### Phylogeny and taxonomy of sobemovirus species

We found some degree of clustering between the phylogenetic tree of 26 sobemovirus species (see **Methods**; see also **Table 1**) and that of their respective plant family hosts (**Fig 2A**). The eight sobemovirus species of monocots clustered in two distinct lineages, one consisting of five virus species infecting host plants of the Poaceae family, including RYMV (referred to below as the RYMV lineage), and another consisting of three virus species infecting plants of the Poales order, including ryegrass mottle virus (RGMoV). The seven species infecting plants of the Fabaceae family are split into two lineages, including five and two species. The three species infecting plants of the Solanaceae family belong to a single lineage. The remaining eight virus species infect plants belonging to different and unique families (see **Fig 2A**). In addition to

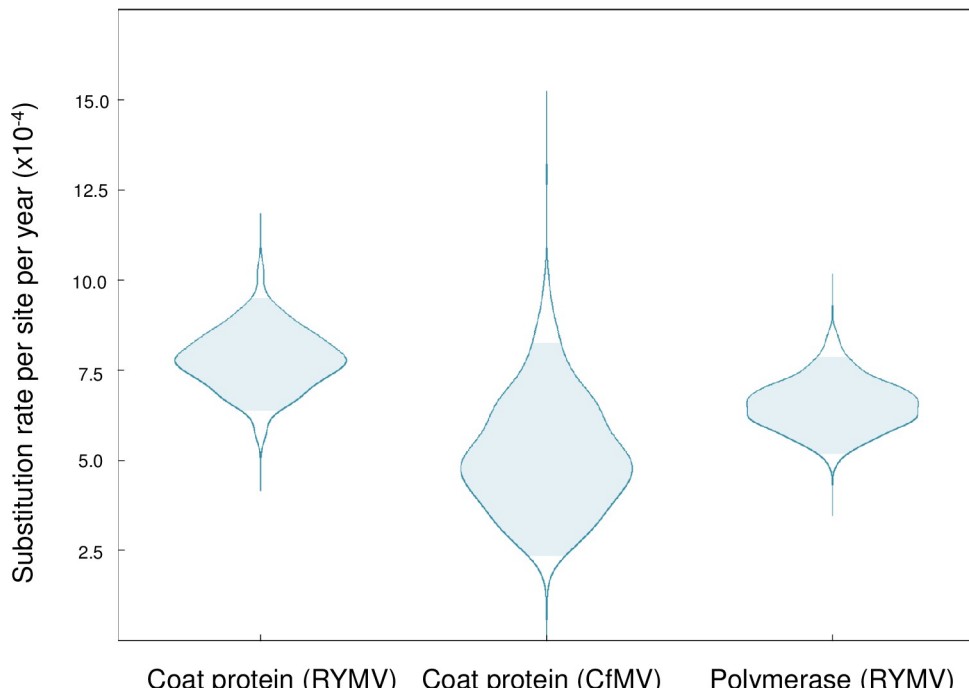

**Fig 1. Posterior substitution rates (number of substitutions per site per year) of the rice yellow mottle virus (RYMV) and cocksfoot mottle virus (CfMV) datasets.** The RYMV dataset has a stronger clock signal and lower variance compared to the CfMV dataset. The shaded area in light blue shows the 95% highest posterior density region.

evidence of higher barriers to infection between plant families, we see similar evidence at the within-family rank, as exemplified by the RYMV lineage. Collectively, the species of the RYMV lineage infect plants of several genera in the Poaceae family, but individually, they have a narrow host range that is sometimes constrained to one or a few genera. The sister species CfMV and cynosurus mottle virus (CnMoV), which both originated in Europe, have an overlapping host range, both infecting the temperate crops wheat, oat, barley, and rye but not maize, sorghum or rice [31]. RYMV infects plants belonging to the *Oryza* genus whereas its sister species, imperata yellow mottle virus (IYMV), found exclusively in Africa, infects only maize, *Imperata cylindrica*, and *Rottboellia exaltata* [32].

The distribution of pairwise genetic identity in ORF2b across the sobemovirus species revealed a major mode from 57 to 73% genetic identity, a minor mode from 77 to 84%, and a "shoulder" from 91 to 100% (**Fig 2A**). This showed that nearly every pair of the sobemovirus species have 57–73% genetic identity with the exception of the CnMoV-Poaceae Liege sobemovirus (PLSV) pair and pairs among southern bean mosaic virus (SBMV), sesbania mosaic virus (SeMV) and soybean yellow common mosaic virus (SYCMV) trio. These groups, hereafter referred to as subspecies in the *Sobemovirus* genus, have intermediate levels of genetic identity (77–84%). Finally, the shoulder reflects the intraspecific genetic identity between pairs of isolates, ranging from 90.6 to 98.6%, with RYMV having a 92.1% minimum pairwise intraspecific identity (**Fig 2A**; see also **S2 Table**).

## Phylogeny and taxonomy of recognized and tentatively new sobemovirus species

We next examined a dataset of the 26 sobemovirus species isolated from plants (examined above) merged with 42 additional known or tentatively novel sobemovirus sequences collected

**Table 1. Virus name, family host, and accession numbers of the 26 sobemovirus species and candidate species.**

| Virus | | Host | Accession number |
|---|---|---|---|
| Artemisia virus A | (ArtVA) | Asteraceae | NC017914 |
| Blueberry shoestring virus | (BSSV) | Ericaceae | LC081344*** |
| Cocksfoot mottle virus | (CfMV) | Poaceae | DQ680848 |
| Cymbidium chlorotic mosaic virus | (CyCMV) | Orchidaceae | LC381945 |
| Cynosorus mottle virus* | (CnMoV) | Poaceae | OM323994 |
| Imperata yellow mottle virus | (IYMV) | Poaceae | NC011536 |
| Lucerne transient streak virus | (LTSV) | Fabaceae | NC001696 |
| Mimosa mosaic virus* | (MimMV) | Fabaceae | OP456085*** |
| Papaya lethal yellowing virus | (PLYV) | Caricaceae | NC018449 |
| Physalis rugose mosaic virus | (PhyRMV) | Solanaceae | MK681145*** |
| Pistacia sobemovirus* | (PisSV) | Anacardiaceae | MT334602*** |
| Poaceae Liege sobemovirus* | (PLSV) | Poaceae | ON137710*** |
| Rice yellow mottle virus** | (RYMV) | Poaceae | AJ608207 |
| Rottboellia yellow mottle virus | (RoMoV) | Poaceae | KC577469 |
| Ryegrass mottle virus | (RGMoV) | Poaceae | EF091714 |
| Sesbania mosaic virus | (SeMV) | Fabaceae | NC002568 |
| Snake melon asteroid mosaic virus | (SMAMV) | Cucurbitaceae | MT989351*** |
| Solanum nodiflorum mottle virus | (SNMoV) | Solanaceae | NC033706 |
| Southern bean mosaic virus | (SBMV) | Fabaceae | AF055887 |
| Southern cowpea mosaic virus | (SCPMV) | Fabaceae | NC001625 |
| Sowbane mosaic virus | (SoMV) | Amaranthaceae | GQ845002 |
| Soybean yellow common mosaic virus | (SYCMV) | Fabaceae | KX096577 |
| Subterranean clover mottle virus | (SCMoV) | Fabaceae | AY376451 |
| Turnip rosette virus | (TRoV) | Brassicaceae | NC004553 |
| Velvet tobacco mottle virus | (VTMoV) | Solanaceae | NC014509 |
| Xufa yellow dwarf virus* | (XYDV) | Cyperaceae | ON828429*** |

*candidate species, proposed to ICTV to be recognized as novel species in 2023

**the accession number for two most divergent pairs of RYMV sequences selected for the PoW analysis are MF989228 (RYMV-1) and MZ172959 (RYMV-2).

***viral sequences assembled from high-throughput sequencing of plant viromes.

from metagenomic analyses (hereafter referred to as metagenomic sequences; see **Methods**) to establish their taxonomy and phylogenetic relationships. Most sequences were obtained from gut metagenomes of herbivorous animals feeding mainly on Poaceae and Fabaceae (**Table 2**). This explains why many isolates were related to sobemovirus species found in Fabaceae and in Poaceae, particularly RGMoV (see **Fig 2B**). In contrast, none of the metagenomic sequences clustered closely with sobemovirus species infecting the Solanaceae (see **Fig 2B** and **Table 2**). We found that although the great majority of the metagenomic sequences originated from China [33], they mostly cluster with isolates found in other continents, reflecting the world-wide distribution of many sobemoviruses. Nevertheless, there was some geographical association between the sequences. For instance, the metagenomic sequences close to CfMV (MN626425 and MW588040) and CnMoV (MW588173) were from Europe.

The phylogeny for the collection of 68 sequences differed in several respects from that of the 26 selected samples representing the recognized and candidate sobemovirus species that we initially investigated. The overall genetic diversity of the merged dataset was noticeably higher as new subclades possibly representing the novel species were revealed, some of which have long independent evolutionary histories (see **Fig 2B**). There were also several sequences

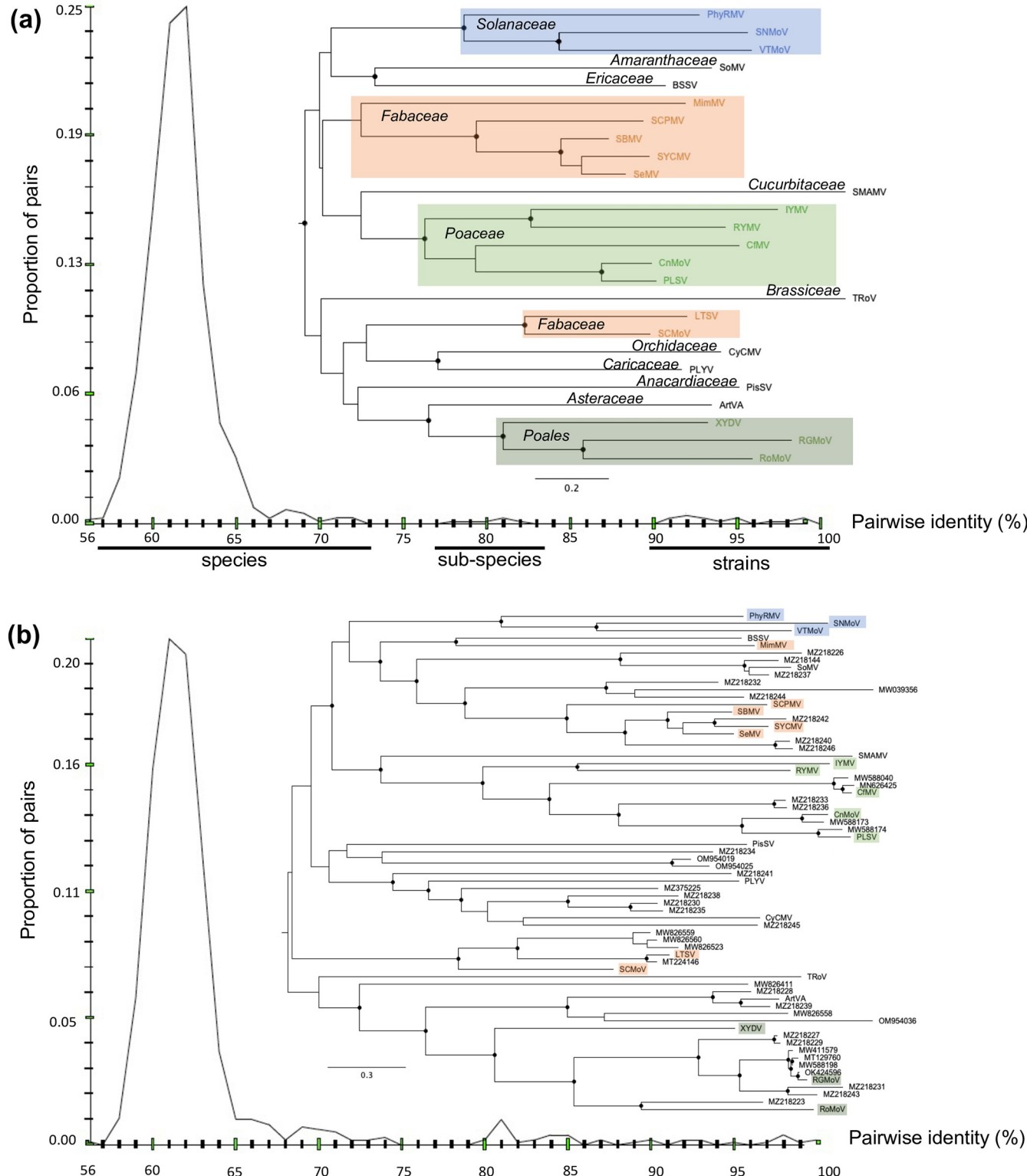

**Fig 2. Phylogeny and pairwise distribution of genetic identity of known and tentative sobemovirus sequences identified in metagenomic analysis of the ORF2b (polymerase) region. (a)** Phylogeny and pairwise identity of the 26 sobemovirus species. The host plant family of each species is highlighted on the tree. For 12 (out of 26) species with at least two isolates (see S2 Table), the two most divergent isolates were compared for pairwise genetic identity. The species, subspecies, and strain tentative taxonomical levels are indicated at the bottom of the figure. **(b)** Phylogeny and pairwise identity of the 26 sequences representing recognized sobemovirus species and 42 sequences of tentative sobemovirus isolates originating from metagenomic studies. The virus species

infecting plants belonging to the Solanaceae, Fabaceae, Poaceae families, or the Poales order are indicated in blue, orange, light green, and dark green, respectively. The accession numbers for the 42 additional sequences are shown at the tips of the tree. Black circles show internal nodes with at least 90% bootstrap supports.

that are closely related to known sobemovirus species such as RGMoV. The RGMoV lineage includes strains of RGMoV and related species, likely at the subspecies level. Our analysis of the pairwise genetic identity of the merged dataset revealed a continuous change in pairwise identity of samples from interspecific to intraspecific levels. This contrasts with the earlier findings in the 26 selected samples of sobemovirus species where certain percentages of genetic identity had no corresponding pairs (see **Fig 2A**). This discontinuity is no longer apparent in the merged dataset, confirming the broadened sobemovirus genetic diversity (**Fig 2B**).

## Deep evolutionary history of sobemoviruses

We first constructed a time-scaled phylogenetic history for the 26 sobemoviruses using the PoW model (see **Methods**). This analysis dated the most recent common ancestor (TMRCA) of known sobemoviruses back to 3.4 (95% HPD: 1.8–5.4) million years before present (BP) (**Fig 3A**), much earlier than previous estimates based on extrapolation of short-term substitution rate of RYMV [23]. The deep evolutionary history of sobemoviruses may indicate a long-term association of these viruses with their host, with host jumps that occurred over long evolutionary timescales. For instance, we estimated the TMRCA of virus species infecting plants of the Poaceae, Fabaceae and Solanaceae at 546,000 (95% HPD: 260,000–1,490,000), 90,000 (30,000–186,000), and 299,000 (109,000–616,000) years BP, respectively. Within the RYMV lineage, the RYMV/IYMV split was approximately 117,000 years BP while the CfMV/CnMoV split was close to 151,000 years BP. Similarly, the TMRCA of most species was between several dozens of thousands to a few hundred thousands of years BP, with turnip rosette virus (TRoV) and snake melon asteroid mosaic virus (SMAMV) diverging for over a million years before sharing an ancestry with another sobemovirus (**Fig 3A**). The notable exceptions came from the subspecies identified in our taxonomy analysis. The SeMV/SYCMV and CnMoV/PLSV splits were estimated to approximately 2,600 and 3,000 years BP, respectively, while the MRCA of the SBMV, SeMV, and SYCMV existed approximately 9,000 years BP. Over shorter timescales, we found that the TMRCA of the two most divergent RYMV isolates (RYMV-1 and -2 in **Fig 3A**) was 156 (95% HPD: 80–288) years BP, in agreement with previous estimates obtained from standard Bayesian molecular clock analysis which do not correct for TDRP [34]. This finding further suggests that saturation over such short timescales is unlikely to have a significant impact on TMRCA estimates of sobemoviruses.

We next constructed the PoW-transformed time tree of the 68 sobemovirus sequences (**Fig 3B**) and found that the root of the tree is 4.3 (95% HPD: 2.0–6.6) million years BP, largely similar to the estimates obtained with the 26 sobemovirus dataset. Interestingly, we see that many of the sequences of metagenomic origin share common ancestors with closely related known sobemovirus species (i.e., they belong to the same species but may represent new subspecies or strains) and diverged less than 10,000 years BP. This timeline suggests major radiation events aligning with the Neolithic period, which varied in timing across different geographical regions. To investigate this statement, we constructed a lineages-through-time plot, which shows a sharp increase in the number of diversification events between 4,600 to 2,000 years ago (see **Fig 3B**). In particular, the TMRCA of the RGMoV lineage (consisting of strains of RGMoV and closely related tentative unrecognized species) was approximately 4,600 years BP. Other radiation events such as those between sobemovirus and metagenomic sequences (RoMoV/MZ218223 split 2,300 years BP) or between metagenomic sequences (MZ218230,

**Table 2. Isolation source and Genbank accession numbers of the 42 sequences obtained from non-plant metagenomic analysis.**

| Gut metagenome | Accession number |
|---|---|
| lizard | MZ375225 |
| bird | MW826411 |
| bird | MW826523 |
| chicken | MZ218230 |
| swan | MW588040 |
| swan | MW588173 |
| swan | MW588174 |
| swan | MW588198 |
| cattle | MZ218223 |
| cattle | MZ218227 |
| cattle | MZ218229 |
| cattle | MZ218234 |
| cattle | MZ218239 |
| cattle | MZ218242 |
| cattle | MZ218246 |
| horse | MZ218231 |
| pig | MZ218144 |
| rabbit | MT129760 |
| rabbit | MN626425 |
| rat | MW826558 |
| rat | MW826559 |
| rat | MW826560 |
| sheep | MZ218226 |
| sheep | MZ218233 |
| sheep | MZ218236 |
| sheep | MZ218237 |
| sheep | MZ218240 |
| sheep | MZ218244 |
| Homo sapiens | OK424596 |
| **Sediment metagenome** | |
| semi-desert pond | MZ218228 |
| fishpond | MZ218232 |
| reservoir | MZ218238 |
| sediment | MZ218243 |
| river water | OM954019 |
| river water | OM954025 |
| river water | OM954036 |
| irrigation water | MW411579 |
| **Insect metagenome** | |
| soybean thrips | MW039356 |
| soybean thrips | MT224146 |
| **Soil metagenome** | |
| old-growth forest | MZ218241 |
| old-growth forest | MZ218245 |
| forest | MZ218235 |

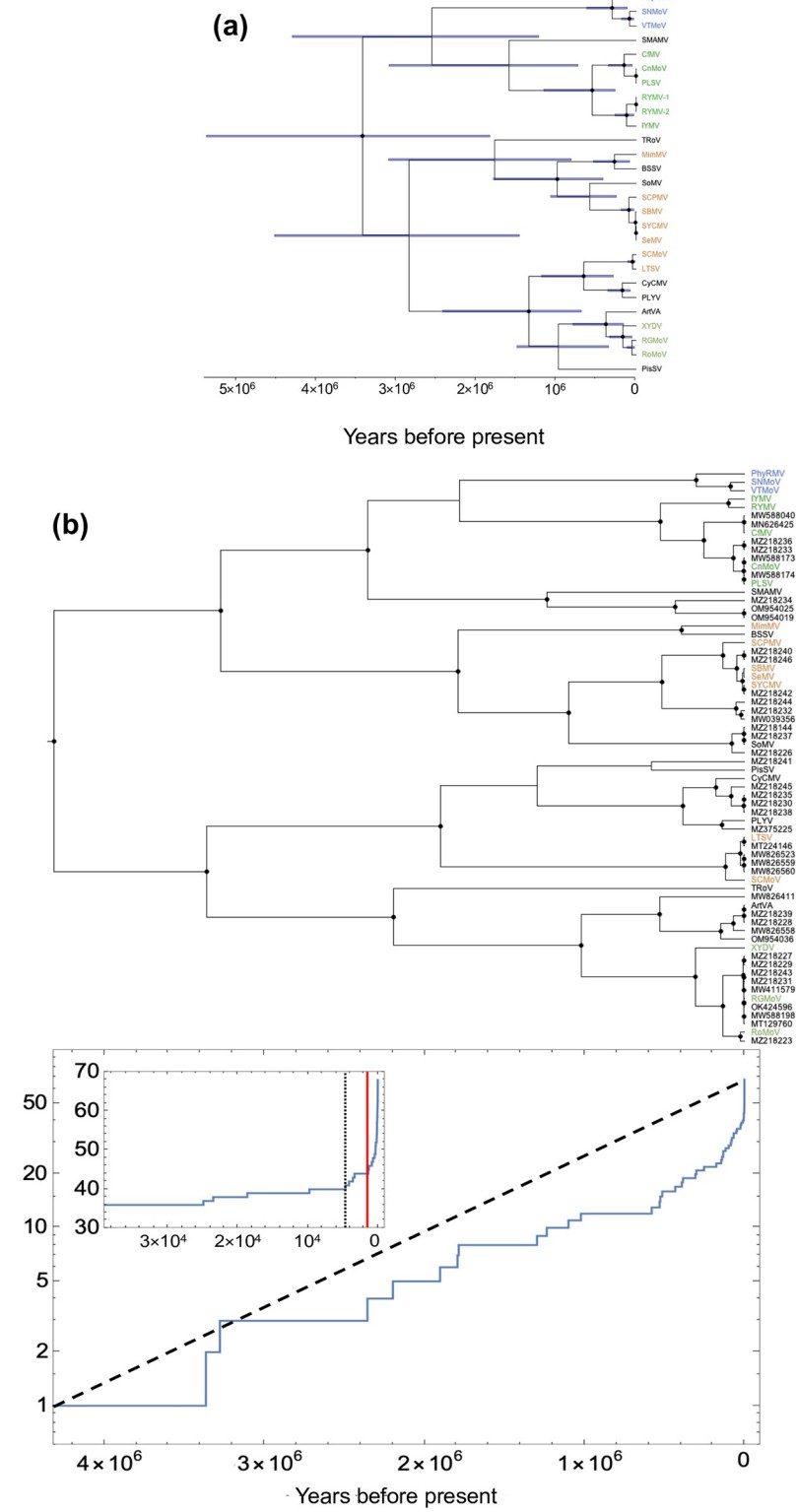

**Fig 3. PoW-transformed time tree of the sobemovirus species and number of diversification events through time.**
(**a**) PoW-transformed time tree of 26 sobemovirus sequences. RYMV is represented by the sequences of its two most divergent isolates MF989228 (RYMV-1) and MZ172959 (RYMV-2). The horizontal blue bars show 95% highest posterior density of the node age. (**b**) PoW-transformed time tree of 68 known and tentative sobemovirus sequences (top panel) and their respective number of diversification events through time (bottom panel). Black dashed line shows

a constant rate of speciation event over time with no extinction. The inset shows the number of diversification events over the last 40,000 years. Black vertical dotted line indicates 4,600 years ago, a point beyond which nearly 40% of diversifications occurred until present. Red vertical line indicates 2,000 years ago, a point on the graph with the sharpest increase in the number of diversification events. The virus species infecting plants belonging to the Solanaceae, Fabaceae, Poaceae families, or the Poales order are indicated in blue, orange, light green, and dark green, respectively. The accession numbers for the 42 additional sequences are shown at the tips of the tree. Black circles show internal nodes with at least 90% posterior supports.

MZ218235, and MZ218238 split 3,500 years BP) occurred within the last few thousand years. Several divergence events were estimated to have taken place less than 200 years ago, corresponding to pairwise genetic identity higher than 90% between sequences reflecting divergence among strains.

## Discussion

Using an evolutionary model that accounts for time-dependent changes in the evolutionary rates over time, we were able to show that the *Sobemovirus* genus is a few million years old, making a case for ancient origins of plant viruses. Our analysis revealed that the species in this genus became increasingly specialised through time, with likely a few deep host jumps to members of the Poaceae, Fabaceae, and Solanaceae families that occurred tens to hundreds of thousand years ago. The association between the phylogeny of the sobemovirus species and their host range is consistent with the idea that the taxonomic rank of the host family sets a threshold for plant-virus host range, with a higher barrier to infection between plant families [35]. Crucially, our estimates for the evolutionary history of sobemoviruses are several orders of magnitude older than the Neolithic, a result that is in stark contrast with previous estimates that suggested divergence time between sobemoviruses and related viruses is only 9,000 years old [23]. However, we did find evidence of major diversification during the Neolithic, partly supporting the hypothesis that initial radiation of several plant virus groups, including the *Potyvirus* genus, took place at the dawn of agriculture [20,23].

While we found that TMRCA of RYMV using the PoW model is very similar to previous estimates using its short-term substitution rate, extrapolations of this rate to infer timescale of evolutionary events within the *Sobemovirus* genus resulted in a significant underestimation of the evolutionary history of sobemovirus species. The PoW analysis also revealed a clear association between divergence dates and taxonomic levels: sequences with a genetic identity below 75% (species level) trace back to tens of thousands of years; those between 75% and 90% (subspecies level) emerged within the Neolithic era, thousand years ago. Meanwhile, sequences with over 90% identity (intraspecific level) indicated a recent divergence of one to two hundred years ago (strain level).

Thanks to high particle stability of sobemovirus virions in the environment [27], we were able to include the samples collected through metagenomic analysis of the phytobiome and investigated their evolutionary history along with known and candidate sobemovirus species. This analysis revealed an even wider diversity of sobemovirus species at the interspecific and intraspecific levels. Overall, our findings underscored the critical impact of TDRP in estimating long-term evolutionary history of viruses and offered a new perspective on the possible long-term virus-host associations within the *Sobemovirus* genus. This is exemplified by host range evolution, a pivotal aspect of plant virus emergence that has garnered significant attention in recent years (see [36,37] for recent reviews).

Our findings confirmed that the adaptation of RYMV to the African rice *Oryza glaberrima* occurred during the diversification of the virus in West-Africa, within the past two hundred years [38]. The association between the geographical distribution of virus species and their

respective host plant families, such as the relationships observed between CfMV and CnMoV with temperate crops, or RYMV and IYMV with tropical crops, may reflect historical shifts in virus host preferences during the course of agricultural development, as has been postulated for potyviruses [39]. However, adaptation to wild species in the *Oryza* genus may have occurred much earlier, independently from human intervention with rice domestication or rice cultivation, considering that RYMV and IYMV diverged approximately 117,000 years ago. While our findings of the deep history of the *Sobemovirus* genus expands the range of time-scales involved in the evolution of the plant virus host range by hundreds of thousand years, it precludes host-virus co-speciation scenarios. For instance, *Oryza sativa* and *Imperata cylindrica*, the respective hosts of RYMV and IYMV, diverged approximately 59 million years ago [40], much earlier than our estimate for the RYMV/IYMV split 117,000 years ago.

In interpreting our results, it is important to acknowledge the inherent uncertainties associated with reconstructing time trees, particularly those related to deep evolutionary histories, as evident in the large confidence intervals from **Fig 3A**. These uncertainties are an intrinsic aspect of reconstructing the deep evolutionary history of viruses where our estimates are increasingly susceptible to the effects of site saturation on inferred divergence times from a common ancestor. Such limitations, while challenging, are commonplace in studies of historical nature, particularly because of the scarcity of direct evidence. Our study, therefore, presented a model that is grounded in the best available data and methodologies, but also one that is open to refinement as new information emerges.

We specifically applied the PoW model to sobemoviruses. This focused approach is particularly important in instances where the evolutionary rates within a virus family or genus do not show significant variance across different lineages. One of the challenges of expanding this model to other virus families apart from constructing a reliable dataset to infer short-term substitution rate of at least more than one virus lineage within a genus, is difficulties associated with aligning highly divergent sequences at the family level and beyond. These challenges are rooted in the limitations of current sequence alignment methods rather than the PoW model itself.

For future work, there are promising avenues for investigation using the PoW model. For instance, exploring the evolutionary history at the family and interfamily levels, particularly between families like *Solemoviridae* and *Barnaviridae*, could yield insight on the evolutionary origins of plant and fungus viruses. Another area of interest lies in the study of *Polerovirus*, a sister genus of *Sobemovirus* genus within the *Solemoviridae* family. Investigating *Polerovirus* genus could provide valuable contrasts, given its different ecological significance and broader species range.

These potential research directions, however, require tailored applications of the PoW model, ensuring accurate evolutionary interpretations. Our focus on sobemoviruses in this work sets the foundation for these future endeavors. Expanding knowledge on the deep history of plant viruses is essential to uncovering the ways in which they have been influenced by the development of agriculture and other environmental perturbations, including climate change [41]. These findings may also have practical implications for crop protection strategies and for ensuring the sustainability of agricultural systems in the face of evolving viral challenges.

## Methods

### Short-term substitution rate estimation

The sequences of the coat protein gene of 51 isolates of cocksfoot mottle virus (CfMV) collected between 1978 and 2022 (44 years) throughout the world were downloaded from sequence databanks or sequenced in this study (see **S1 Table**). As the 5'end of the sequences of

some of the early isolates was not available, a truncated alignment was made of 640 nt long sequences (instead of 765 nt). We evaluated the strength of the temporal signal in the CfMV dataset using a tip cluster-randomization test in root-to-tip regression, in addition to conducting a Bayesian assessment of temporal signal within the data [42,43]. Similarly, for rice yellow mottle virus (RYMV), we used a previously published dataset of 261 isolates of the coat protein gene (720 nt) collected in West Africa between 1975 and 2018 (43 years). We found evidence of sufficient temporal signal for rate inference in both CfMV and RYMV datasets under an HKY85+Gamma substitution model and a constant population size coalescent prior in BEAST (v 1.10.4) [44]. The substitution rate of the polymerase of RYMV was estimated from an alignment of 69 dated ORF2b sequences (1515 nt) by calibrating the root with a normally distributed prior with a mean of 160 years and a standard deviation of 20 years, derived from the analysis of the coat protein gene sequences [34].

### Phylogeny and taxonomy of the sobemovirus species

The sobemovirus dataset consists of 26 species, including 21 species recognized by ICTV and five candidate species (see **Table 1**). The sequences were codon-aligned using MUSCLE [45] and the absence of recombination events between species was confirmed with the Phi test [46,47]. We used the ORF2b alignment (encoding the polymerase) as this is the most conserved part of the genome at the genus level to construct the sobemovirus phylogeny under a HKY substitution model with an optimized rate variation using PhyML implemented in Sea-View (v 5.05) [48]. We truncated the 3' end of the ORF2b which had poor coverage across many samples to reach a final alignment length of 1140 nt. For 12 (out of 26) sobemovirus species with at least two isolates available, we further assessed the interspecific range of identity within the ORF2b region by taking the two most divergent isolates of each species using the Sequence Demarcation Tool (SDT2) [49]. The number of isolates within each species varied from 2 to 72 (see **S2 Table**).

### Metagenomic analysis

On August 1, 2023, we retrieved a total of 42 sequences of putative sobemoviruses that were collected as components of the phytobiome from gut, sediment, soil, and insect metagenomes from NCBI (see **Table 2**), reflecting the high stability of the sobemoviruses. We referred to them as "metagenomic" sequences since there was no information on the original host plants they may infect, a common problem when dealing with metagenome data (see [50–52]). The sequences have variable lengths, with several of them having near-complete genome lengths. We then constructed an alignment of the ORF2b region for the 42 metagenomic and 26 sobemovirus sequences with plant-origin representing the known species (see above) for the phylogenetic analysis. The distribution of pairwise genetic identity between all 68 sequences was assessed using SDT2. Similar to the sobemovirus species dataset, the pool of 68 sequences were codon-aligned using MUSCLE, and the absence of recombination events between species was confirmed with a Phi test.

### Time tree reconstruction with PoW model

We used the Prisoner of War (PoW) model of virus evolution to account for the power-law decay in the substitution rate estimates of RYMV over time in our estimation of the evolutionary time scale of the *Sobemovirus* genus [10]. Initially, we took the short-term substitution rate of RYMV's polymerase gene, inferred from a standard molecular clock model (see above), as the baseline rate for sobemoviruses in the absence of the time-dependent rate decay which comes into effect over long timescales. We then constructed ultrametric distance trees for two

datasets: one comprising 26 sobemovirus species and the other an expanded set of 68 sequences, which included sequences of metagenomic origin. These trees were built under the standard HKY substitution model with a strict molecular clock in BEAST 1. These two steps are necessary for establishing the phylogenetic framework for the PoW model (see ref [10] for more details on the PoW model).

The PoW model employs two parameters to transform these ultrametric trees into time trees. The first is the short-term substitution rate for sobemoviruses which we estimated using the time-stamped sequences of RYMV for the polymerase gene. The second parameter is the substitution rate at the fastest-evolving rate group, set at $3.65 \times 10^{-2}$ s/s/y in the model for all RNA viruses. This parameter combination delineates the proportion of sites evolving at various rates, from the fastest-evolving to the slowest (approximating the host rate).

To construct the PoW-transformed time trees, we sampled 100 iterations from the post-burn-in posterior rate distributions of the RYMV substitution rate and ultrametric distance trees, excluding the initial 10% of MCMC chain runs. We finally used TreeAnnotator v.1.10.4 to merge the 100 reconstructed time trees into a maximum clade credibility tree, representing the most probable evolutionary history according to the PoW model.

To visualize the diversification patterns across the Sobemovirus genus, we generated a lineages-through-time plot using the phytools library in R [53] for the complete dataset of 68 sequences. This visual representation allowed us to assess the rate and timing of evolutionary divergence within the genus.

## Supporting information

**S1 Table. Accession numbers and sampling years of the cocksfoot mottle virus isolates.**
(DOCX)

**S2 Table. Percentage of sequence identity in ORF2b between the most divergent pair of isolates within each species.**
(DOCX)

## Acknowledgments

In memoriam: We acknowledge Erkki Truve, who passed away in April 2020. His contribution to the research on sobemoviruses is invaluable and it is thanks to his work that this research continues in Estonia to this day. We remember his dedication and expertise, and his work remains an integral part of this article.

## Author Contributions

**Conceptualization:** Mahan Ghafari, Denis Fargette.

**Data curation:** Merike Sõmera, Cecilia Sarmiento, Annette Niehl.

**Formal analysis:** Mahan Ghafari, Eugénie Hébrard, Philippe Lemey, Denis Fargette.

**Supervision:** Denis Fargette.

**Writing – original draft:** Mahan Ghafari, Denis Fargette.

**Writing – review & editing:** Mahan Ghafari, Merike Sõmera, Cecilia Sarmiento, Annette Niehl, Eugénie Hébrard, Theocharis Tsoleridis, Jonathan Ball, Benoît Moury, Philippe Lemey, Aris Katzourakis, Denis Fargette.

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
