## [Decision Letter · Decision Letter 0]

5 Dec 2023

Dear %TITLE% Ghafari,

Thank you very much for submitting your manuscript "Revisiting the origins of the Sobemovirus genus: a case for ancient origins of plant viruses" for consideration at PLOS Pathogens. As with all papers reviewed by the journal, your manuscript was reviewed by members of the editorial board and by several independent reviewers. The reviewers appreciated the attention to an important topic. Based on the reviews, we are likely to accept this manuscript for publication, providing that you modify the manuscript according to the review recommendations.

Sincerely,

Peter D. Nagy

Academic Editor

PLOS Pathogens

Savithramma Dinesh-Kumar

Section Editor

PLOS Pathogens

Kasturi Haldar

Editor-in-Chief

PLOS Pathogens

orcid.org/0000-0001-5065-158X

Michael Malim

Editor-in-Chief

PLOS Pathogens

orcid.org/0000-0002-7699-2064

Reviewer Comments (if any, and for reference):

Reviewer's Responses to Questions

**Part I - Summary**

Reviewer #1: This very interesting study revises the evolutionary history of sobemoviruses by taking into account the TDRP. As a result it is proposed that the origin of this genus, and its initial diversification on infect different plant families, is orders of magnitude older than previously considered. The study is based in all available information on sobemovirus diversity, and uses the most appropriate methodology, and the paper is well written and easy to read. I have only minor suggestion that could improve the text.

Reviewer #2: In this interesting and potentially important paper, Ghafari et al. estimate the age of the viruses in the family Sobemoviridae, taking into account the time-dependent rate phenomenon using the Prisoner of War model.

The come up with an estimate of about 4 million years for the origin of sobemoviruses, orders of magnitude older than previously believed. They then explore the evolutionary history of this virus family, revealing likely host jumps and major radiation during the Neoltithic period. Such findings are indisputably important for understanding virus evolution. The article is very clearly and well written but, in the opinion of this reviewer, requires a substnatial amendment to become fully convincing, as detailed below.

**Part II – Major Issues: Key Experiments Required for Acceptance**

Reviewer #1: There are no major issues.

Reviewer #2: Surprisingly, nowhere in the paper is the Prisoner of War model actually explained. This seems to be a must for the reader to be able to understand what was actually done. An adequate explanation has to be included in the Introduction. Even more importantly, the description of the model in Methods section is overly brief and effectively uninformative. It needs to be adequately expanded including the statistical evaluation of the results. As it stands, the reader simply has to believe the estimates which is not acceptable.

**Part III – Minor Issues: Editorial and Data Presentation Modifications**

Reviewer #1: 1. Abstract and elsewhere. The authors often refer to the Neolithic as a time frame, which may be confusing and is probably incorrect. The neolithic refers to a period of technical development associated with agriculture, and occurs at different periods in different regions of the world, the 10000 yr PB mentioned is mostly of application to the Near East. As sobemoviruses seem to have “speciated” in different regions of the world (as stated for CfMV and RYMV) the time frame for the neolithic may vary very much. Certainly, neither in West Africa nor in temperate Europe did the neolithic occur 10000 yr ago. It would be thus good to rephrase the corresponding sentences.

2. Lines 111.113. No need to use three independent sentences all starting by “The genome”.

3. Metagenomic sequences. I am surprised that no sequences were retrieved from plant viromes or plant metagenomic data. How is this so? Were plant data bases not searched?, or they were searched and no sobemovirus sequences were found?

4. Line 188. Herbaceous should be substituted by herbivore.

5. Discussion. The uncertainties unavoidably associated to phylogenetic analyses extended back in time, and shown by the huge confidence intervals in Fig. 3a, should be somehow acknowledge in the discussion. Such uncertainties, by the way, do not in the least diminish the interest and novelty of the results, as they are intrinsic to all historic studies.

6. The colours used to indicate host families in Fig. 2b are really hard to see. Perhaps the colours could be used as shadows over the virus name rather than in the letters.

Reviewer #2: I suppose this is too much to ask but to this reviewer, this work would have been far more interesting and compelling if more than one virus family was analyzed and compared. Furthermore, can this analysis be extended to estimate the timeline of virus evolution deeper, beyond the family level? Hopefully, at least, this will be addressed in future work. Perhaps, worth mentioning in the Discussion.

PLOS authors have the option to publish the peer review history of their article (what does this mean?). If published, this will include your full peer review and any attached files.

Reviewer #1: No

Reviewer #2: **Yes: **Eugene V Koonin

Figure Files:

Data Requirements:

Reproducibility:

References:

---

## [Editor Report · Decision Letter 1]

18 Dec 2023

Dear %TITLE% Ghafari,

We are pleased to inform you that your manuscript 'Revisiting the origins of the Sobemovirus genus: a case for ancient origins of plant viruses' has been provisionally accepted for publication in PLOS Pathogens.

Best regards,

Peter D. Nagy

Academic Editor

PLOS Pathogens

Savithramma Dinesh-Kumar

Section Editor

PLOS Pathogens

Kasturi Haldar

Editor-in-Chief

PLOS Pathogens

orcid.org/0000-0001-5065-158X

Michael Malim

Editor-in-Chief

PLOS Pathogens

orcid.org/0000-0002-7699-2064
---

## [Editor Report · Acceptance letter]

8 Jan 2024

Dear Dr Ghafari,

We are delighted to inform you that your manuscript, "Revisiting the origins of the <i>Sobemovirus<i> genus: a case for ancient origins of plant viruses," has been formally accepted for publication in PLOS Pathogens.

Best regards,

Michael Malim

Editor-in-Chief

PLOS Pathogens

orcid.org/0000-0002-7699-2064